# A Meta-Synthesis Study of Person-Centered Care Experience from the Perspective of Nursing Home Residents

**DOI:** 10.3390/ijerph19148576

**Published:** 2022-07-14

**Authors:** Eun-Young Kim, Sung-Ok Chang

**Affiliations:** 1College of Nursing, Korea University, Seoul 02841, Korea; magic0614@korea.ac.kr; 2College of Nursing, and BK21 FOUR R&E Center for Learning Health Systems, Korea University, Seoul 02841, Korea

**Keywords:** nursing homes, person-centered care, systematic review, qualitative research

## Abstract

Purpose: To systematically review and synthesize the evidence for the experience of person-centered care from the perspective of nursing home residents to understand their views in depth. Methods: The seven steps of Nobit and Hare’s meta-ethnography, a well-known meta-synthesis method, were applied. We used four databases for the literature search (PubMed, Web of Science, EMBASE, and CINAHL). Results: A total of seven studies were included for review. As a result of synthesizing the results, three themes (“promotion of mutual understanding through communication”, “care that acknowledges the independence of residents” and “finding the optimized state”) and six sub-themes were derived. Conclusions: This study has provided an in-depth understanding of person-centered care and will contribute to increasing its practical application.

## 1. Introduction

As the world’s population ages, the number of nursing homes (NHs), which act as elderly medical welfare facilities, has risen significantly [1]. An NH can be defined as a place that reflects the identity of the residents as a residential space that has replaced their previous homes [2]. New residents experience a state of psychological instability [3] when they leave their homes to enter an NH and can feel confused about their identity while experiencing extreme stress and psychological changes [4]. Consequently, the quality of care provided to NH residents and the settings in which they live play crucial roles in their adaptation and have a direct impact on their quality of life [4]. NH residents are elderly patients, often with cognitive and chronic diseases, and providing high-quality care to individuals necessitates social attention. As issues such as a lack of care [5], levels of disparity across facilities [6], and elderly abuse [7] have surfaced, so have negative perceptions of the nursing care offered in NHs. In recent years, a paradigm shift in nursing has occurred, from disease-centered nursing, which focused on medical professionals, to person-centered care (PCC), which reflects a patient’s wants and preferences [8]. The concept of PCC is focused on providing personalized nursing care that represents patient choices, values, and requirements [9,10].

It has been asserted that the PCC concept originated with Florence Nightingale, who proposed a type of nursing that focused on the patient rather than on the disease [11]. The person-centered approach was later developed by the American psychologist Carl Rogers [12], and the term patient-centered was coined by Balin in the 1960s [13]. This concept has been studied steadily, and in the 1990s, Stewart et al. suggested that doctors needed to gain an understanding of the patient and the disease or condition that was more human-centered and that the interpersonal relationship between the doctor and the patient, i.e., PCC, was considered to be the most important factor in such an approach [14]. In addition, the concept of PCC implies a treatment that supports realistic health and life goals by guiding all aspects of patient management once individual values and preferences have been derived and expressed [15].

PCC has been constantly emphasized as a method for maintaining care quality. Its importance has been acknowledged internationally, not only in nursing but also in health care in general [8,9,10]. For example, in the United States, the concept has recently begun to be used with patients with dementia in the context of long-term care [16]. Interest in PCC is increasing in Europe as well, where various attempts are being made to instruct and implement PCC [17], while in Japan studies are being conducted to apply this concept to practice [18].

Various qualitative studies should be carried out to define the nature of a concept and to provide a platform for further quantitative investigations into its nature. While relevant qualitative research is being conducted regarding PCC, understanding of the concept is still limited due to differences in results. In order to undertake future research, it is required to first understand the concept from the existing literature, then determine the research’s foundation, and then finally to offer a study direction.

Qualitative meta-synthesis is a research method that explores the meaning, experience, and perspective of participants’ expressions in existing qualitative research [19]. This produces an integrated analysis, reduces duplication of research, and can suggest specific directions for future research [20].

Therefore, this study was conducted to comprehensively understand the experience of PCC from NH residents’ perspectives by analyzing and synthesizing the results of qualitative studies that have explored the experiences of PCC in NHs using the qualitative meta-synthesis method.

This study was conducted with the purpose of systematically reviewing qualitative research related to the PCC experience of NH residents and synthesizing the results to better understand the PCC experienced by NH residents.

## 2. Materials and Methods

### 2.1. Study Design

This study analyzed and synthesized the results of a qualitative study that explored the PCC experience of residents in NHs using the qualitative meta-synthesis method. We used Noblit and Hare’s [21] meta-ethnography, an interpretive approach suitable for integrative analysis and the formation of new interpretations beyond the discovery of individual qualitative studies [21]. The researchers conducted research in compliance with the Enhancing transparency in reporting the synthesis of qualitative research statement (ENTREQ) guidelines [22]. This study has been registered with the International Prospective Register of Systematic Reviews PROSPERO (ID CRD42022319943).

### 2.2. Data Collection

#### 2.2.1. Literature Search and Selection

In this study, we used a traditional systematic literature review method for literature selection. We performed the literature selection process according to PRISMA’s systematic literature protocol. A literature search was conducted in March 2022. We followed the principles of the PRISMA guidelines. Prior to the literature search, the two authors discussed search strategies and databases to find appropriate literature that met the purpose and inclusion criteria of this study. The authors selected the PubMed and EMBASE databases [23], which are considered the most important databases in medical field literature searches and included the Web of Science database to expand the search to the social sciences field to more comprehensively explore the characteristics of the concept. CINHAL, a database considered important in nursing field literature, was also included. The search terms were “Nursing homes [Mesh]”, “Patients [Mesh]”, “Aged [Mesh]”, “Patient-centered care [Mesh]”, and “Qualitative research [Mesh]”, which were transformed into index terms suitable for each database and searched by combining AND and OR. To achieve a broad literature search, the publication year was not specified. The criteria for literature selection in this study were (1) the participants were residents of NHs, (2) the studies were conducted using a qualitative research methodology, (3) the studies were published in English, and (4) the studies were published in peer-reviewed journals. In the case of a mixed-methods study that included a qualitative research method, only the results of the qualitative research were included if they could be clearly extracted. If participants other than NH residents were included in a study’s sample, only literature in which the results regarding resident perspectives were clearly differentiated was included. The criteria for the exclusion of literature were: (1) studies in which the results of qualitative research were not clearly revealed among studies using mixed methods research methodology, (2) studies in which it was impossible to clearly confirm the perspectives of residents, and (3) studies for which the original text could not be found. The first author performed the literature search, and the process of selecting and excluding literature according to inclusion and exclusion criteria was independently performed by the first author and the corresponding author. In the selection process, any disagreements at each stage were resolved through sufficient discussion. After the first search, 1075 articles were found, with 431 articles excluded as duplicates using Endnote. While reading the titles and abstracts, 537 unrelated documents were excluded, and the original texts of 108 documents were reviewed. In this process, the final seven studies were selected by excluding 36 studies that did not meet the qualitative research criterion and excluding 65 studies that did not match the topic (Figure 1).

#### 2.2.2. Quality Appraisal

The Critical Appraisal Skills Program (CASP) checklist consisting of 10 questions was used to evaluate the reliability and rigor of the final seven studies [24]. Two researchers independently evaluated each paper, and any discrepancies in CASP results were resolved by reaching a consensus through sufficient discussion (Table 1). Since CASP is recommended for the purpose of understanding literature, the researchers did not include it in the inclusion and exclusion criteria.

### 2.3. Data Analysis and Synthesis

The method of Noblit and Hare [21] incorporates key concepts of qualitative research by examining the differences and similarities between studies chosen through comparative analysis. In addition, this methodology is a synthetic procedure that explores and synthesizes qualitative research findings. First-order components (participant conversation citations from the original study) and second-order constructs (the author’s conceptual interpretation of the original study) were extracted during the data extraction procedure. This extracted data was then used for analysis and synthesis. This method allows researchers to grasp new perspectives and essential meanings of phenomena of interest and to develop a deeper understanding of those phenomena by comparing, contrasting, merging, and synthesizing different studies. The selected studies were arranged in this study in order of publication year, from oldest to most recent. Researchers compared concepts found in the oldest study with those found in the next oldest study, and then extracted similar themes and concepts. By repeating this process, broad concepts were reduced into specific categories. The original data were then reanalyzed to evaluate the confirmed findings, and the reviewed findings were then combined into a common concept to produce a theme (third-order construct) that can be deemed a higher-level concept. The two authors continued to discuss their differences of opinion based on their different academic and clinical backgrounds throughout this analysis and synthesis. We selected the most appropriate quotes for each subtopic and included them in the results.

## 3. Results

Finally, seven studies were included in the analysis, and as a result of the CASP quality appraisal, three studies scored 70%, three scored 80%, and one scored 90% (Table 1). Among the selected studies, six were qualitative studies and one was a mixed-methods study that included a qualitative study; the studies were published from 2011 to 2020 (Table 2). The countries where the studies were conducted were Norway, Canada, the United States, the Netherlands, and Germany. The number of residents who participated in the studies was 682, and the age of the participants comprised a wide range, from 52 to 101 years old. Data collection was performed either by focus group interviews or individual interviews. As a result of analyzing and synthesizing the seven studies, the final four themes and eight sub-themes were derived (Table 3).

### 3.1. Theme I. Promotion of Mutual Understanding through Communication

This theme consists of two sub-themes: “building closeness with nursing staff” and “sharing enough information about caring.” NH residents could make their opinions heard by communicating with nursing staff. This helped improve the shared understanding between the residents and staff and enabled the staff to employ a more individualized approach to providing care to patients. Based on this mutual understanding, PCC was performed.

#### 3.1.1. Sub-Theme 1. Building Closeness with Nursing Staff

Residents shared their daily routines with nursing staff [25,26,27,28,29] and developed an intimate relationship while consulting with them about their concerns [25,26,27,28,29]. The residents communicated with the nursing staff in positive language [25,27,28], and the staff tried to clearly understand a resident’s position by expressing it in the resident’s words [16,17,18,19,20]. Through such communication, the residents’ good relationships with the nursing staff were extended to their families [28,30].


*“My doctor still doesn’t want me to go right now because of my heart. After she talked to me about it, I really don’t want to go home, get sick, and have to come back. I was frustrated, but now I’m satisfied with what she’s saying.”*
([28], p. 197)

#### 3.1.2. Sub-Theme 2. Sharing Enough Information about Caring

By sharing information, residents could potentially receive better care. Residents wanted to obtain information about the treatment and plan [28,29], and they were willing to provide their own personal information that could be helpful to nursing staff for their care [29,31]. Residents wanted to be cared for by high-quality medical personnel with medical knowledge [28,29], and they expected a health care provider with detailed knowledge about their condition [29,31].


*“They listen to us more. They ask our opinions. The administrator comes to the resident council meetings. Before it was only the activity director and we wondered if our suggestions got passed on because I never really saw any changes, but now the head guy comes and the director of dining comes. I think that we are being heard and that they are taking us more seriously now.”*
([27], p. 9)


*“Since I’ve been here, my administrator told me not to be afraid to talk to anyone. So of course I go and talk to her if I have a problem. And we sit down and we discuss it, and she takes the action that needs to be taken.”*
([28], p. 193)

### 3.2. Theme II. Care That Acknowledges the Independence of Residents

Residents have their independence recognized by nursing staff who understand their preferences and needs. Their tastes and needs were immediately reflected. This means that they can lead their own lives as independently as possible and focus more on their own lives.

#### 3.2.1. Sub-Theme 3. Respect for the Individuality of Residents

Nursing staff who were aware of the residents’ health conditions [25,29,31] provided care tailored to those residents’ needs [25,27,29,31]. Residents hoped to receive constant care from dedicated nursing staff assigned to them [23,25,31], and they experienced quick responses to their requests from nursing staff [25,26,28,29].


*“Well the challenge is that, depending on staffing level, I mean you can’t get everybody cared for all at the same time. You have to be very aware of knowing your resident, knowing a time that works well for them as well as knowing a time that works well for the congregate environment. And that comes with a skilled worker.”*
([31], p. 10)


*“Well yes, you see—they do the best they can for you. They work hard all day long. They help you right away… if you need it.”*
([25], p. 1361)


*“I rang the bell for help yesterday, and it took 45 min for someone to come help me. When you are old and on diuretics and have to go to the bathroom, you can’t wait that long. I can’t get to the toilet by myself, so I rang again and then finally just had to go. When she got there, the aid was frustrated that I was wet and so was I.”*
([26], p. 708)

#### 3.2.2. Sub-Theme 4. Focusing on the Residents’ Own Life

Having the power to carry out activities of daily living on their own was considered a very important ability for residents of NHs [25,27,28,29,30,31]. Being able to live independent lives without the help of others [25,27,28,29,30,31], to make decisions about one’s own life [16,18,19,20,21,22], and to maintain hobbies [25,26] means that they are masters of their lives. It was about focusing on their own lives.


*“I decide over my own schedule, I’m independent and that is a good feeling. I feel free, and I am too. But of course I am dependent. And that is a feeling of safety…I am safe, you know. My life is so good. I make my own decisions.”*
([25], p. 1362)


*“I forget my problems by reading or looking at something worthwhile on the TV or by being around friends…Going to activities… I like something to do… to be with a crowd… and getting involved in groups.”*
([26], p. 709)

### 3.3. Theme III. Finding the Optimized State

Residents were provided with nursing care to maintain their physical and psychological well-being, and a therapeutic environment was created to help maintain their health. Residents were able to maintain a stable state while maintaining a state of physical, emotional, and environmental optimization.

#### 3.3.1. Sub-Theme 5. Care to Maintain Physical and Psychological Well-Being

NH residents are mostly elderly people, and they want to be provided with care that helps them maintain their physical and psychological well-being. As such, they were provided with nursing care to maintain their physical functions [25,28,29,30,31]. In addition, they valued being able to maintain a healthy normal daily life [25,27,31] and wanted help to live a physically pain-free life [25,29,31]. For this, they required careful monitoring of their current condition to keep it from worsening [25,28,29,30,31], actions for which the nursing staff experienced a positive attitude [25,27,28,31]. Such care provided the residents with physical and psychological stability.


*“Quality of care is related to emotions and experiences in all phases of the disease.”*
([29], p. 6)

#### 3.3.2. Sub-Theme 6. Creating a Therapeutic Environment for Care

In NHs, residents wanted to be guaranteed an environment in which their safety could be guaranteed [25,26,27,29]. Creating a home-like feel made residents feel comfortable and cared for [23,27,31]. In addition, they were able to continue a stable life by being provided with an environment in which their privacy was respected [27,29].


*“It means that this is our home. You have to respect people when you go into their home. You have to knock and ask to come in, not just barge in the door.”*
([27], p. 8)


*“The last time you were here, there were just rules, rules, rules. Now, we have more freedom. No one wakes me up for breakfast anymore at 5:30 a.m. because I asked them not to. They had me on the list before, I don’t know why, but now I can say no. I like that.”*
([28], p. 198)

Holistically, PCC from the resident’s point of view can be described as three processes. First, residents tried to understand each other and NH nursing staff based on mutual trust through communication with nursing staff. Second, care was provided in which the independence of the residents was recognized. Third, through this nursing process, a method for optimizing the physical and psychological well-being of residents was sought by nursing staff. PCC from the resident’s point of view could be comprehensively expressed as “nursing that respects the residents’ independent existence.”

## 4. Discussion

As the proportion of the elderly population continues to increase worldwide, the number of elderly residents entering NHs is increasing [1]. According to these changes, PCC is being emphasized to reduce the negative impacts on residents of leaving their homes and to improve their quality of life. The concept of person-centered care originated from psychology in the 1960s but was not applied in nursing science until relatively recently, after 2010 [32]. In line with this trend, the publication years of the studies included in our meta-synthesis were 2011–2020. The seven studies were conducted in the United States, Norway, Canada, Netherlands, and Germany, i.e., relatively few countries.

Traditionally, NHs have been perceived as lacking individualized programs, living conditions, and care for patients [33]. However, as a result of this study, from the perspective of PCC, the NH was found to be a space where residents were independent and lived their lives to the best of their abilities.

Our meta-synthesis found that communication in NHs has been shown in previous studies to have various positive effects, such as building empathy between residents and nursing staff and helping to improve safety issues that may arise in NHs [34]. However, the reality is that most patients have difficulty expressing their opinions because, due to dementia and cognitive impairment, they have great difficulty communicating [35]. In order to manage this problem, intervention programs using various media to improve the communication ability of residents need to be developed. The present study also confirmed that residents wanted to share care-related information with medical staff. Because it is a reality that most NHs are managed unilaterally, based on the results of this study, efforts should be made at the institutional level to make information related to residents’ care and care plans more accessible to them.

The results of this study showed that residents wanted to be respected and desired to live independently. These results can be seen as reflective of the negative perception that residents in NHs lead very dependent lives and that they live according to instructions without having their opinions and preferences respected [36]. It has been continuously pointed out that care in NHs is provided in a health care provider-centered environment rather than in a person-centered one [37]. To counter this, various measures for enhancing the self-reliance of residents should be considered, including creating an environment in which PCC is provided.

In addition, it was found that residents wanted to acquire a sense of security by receiving physically, emotionally, and environmentally optimized and individualized care. Various efforts are required to make the subjects feel “at home” in NHs [37]. Because NH residents have various kinds of diseases, cognitive levels, and activities of daily living abilities, research on ways to provide care by classifying the physical and emotional stages of residents in order to provide individualized care to residents should be conducted. Nursing staff should have sufficient knowledge of and skills in the individual care to be provided to residents. It requires the efforts of managers as well as nursing staff to provide PCC to residents. Considering that PCC is a concept that has a positive effect on residents, families, and nursing staff, education on PCC should be continuously developed by diversifying the target population for it.

A limitation of this study is that it is difficult to generalize its results internationally because the selected studies were all conducted in Western countries. In addition, since the selected studies were conducted in countries with very different medical systems, the NH situations, level of care, and characteristics of the aged population of each country were not reflected. Therefore, while this study is meaningful in providing an understanding of PCC in a general NH environment, it is necessary to confirm its findings by conducting additional studies in countries in different regions of the world, especially in Asia, where the number of elderly residing in NHs is rapidly increasing.

## 5. Conclusions

In order for NH residents to feel at home in their new residences, care must be provided that respects the patient as an independent entity and that considers individual preferences. This study suggested a direction for future research so that PCC can be provided in NHs. Based on this study, more and varied qualitative research to understand PCC in NHs is required, and analysis should be undertaken from the perspective of not only residents but also nursing staff and family members. Through such multifaceted research, it will be possible to increase the understanding of PCC, which will contribute to increasing its practical application.

## Figures and Tables

**Figure 1 ijerph-19-08576-f001:**
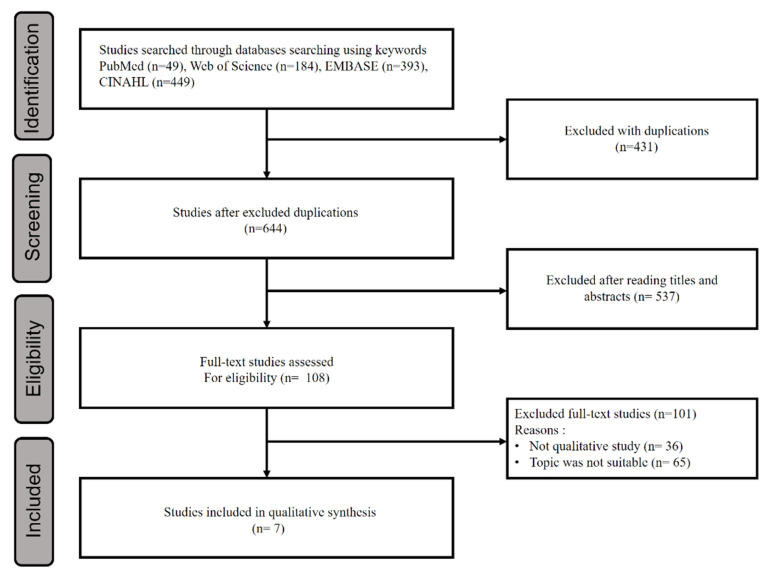
Flow chart of the systematic review used in this study.

**Table 1 ijerph-19-08576-t001:** Quality assessment results of the Critical Appraisal Screening Program using the included studies.

Items	Nakrem et al. (2011) [25]	Bangerter et al. (2016) [26]	Harrison et al. (2017) [27]	Scales et al. (2019) [28]	Vennedey et al. (2020) [29]	Sims-Gould et al. (2014) [30]	Kusmaul et al. (2020) [31]
1. Was there a clear statement of the aims of the research?	Y	N	Y	Y	Y	Y	Y
2. Is a qualitative methodology appropriate?	Y	Y	Y	Y	Y	Y	Y
3. Was the research design appropriate to address the aims of the research?	Y	Y	Y	Y	Y	Y	N
4. Was the recruitment strategy appropriate to the aims of the research?	Y	N	N	N	N	N	Y
5. Was the data collected in a way that addressed the research issue?	N	Y	Y	Y	Y	Y	N
6. Has the relationship between researcher and participants been adequately considered?	Y	Y	N	Y	Y	Y	Y
7. Have ethical issues been taken into consideration?	Y	Y	Y	Y	Y	Y	Y
8. Was the data analysis sufficiently rigorous?	Y	N	N	Y	N	N	N
9. Is there a clear statement of findings?	Y	Y	Y	N	Y	Y	Y
10. Was this research valuable?	Y	Y	Y	Y	Y	Y	Y
Percentage	90%	70%	70%	80%	80%	80%	70%

Y = Yes; N = No.

**Table 2 ijerph-19-08576-t002:** Summary of the included studies.

Author, (Year) [Article No.] /Country	Research Type	Aims	Sample Size (F:M)	Age of Participants (Years)	Data Collection	Data Analysis	Key Findings
Nakrem et al. (2011) [25]/Norway	Quality inquiry study	To describe the nursing home residents’ experience with direct nursing care, related to the interpersonal aspects of quality of care	15 (9:6)	70s: 4 80s: 9 90s: 2	In-depth interview	Content analysis	Care for and alleviation of medical, physical, and psychological needs: general and specialized care, health promotion and prevention of complications, too old and sick to be prioritized? Protecting the residents’ integrity: Self-determination and dependency, altered role from homeowner to resident, fear of indignity and depreciation of social status Psychosocial well-being: balancing the need for social contact and to be alone, preserving the social network
Bangerter et al. (2016) [26]/USA	Qualitative study	To assess older adults’ preferences for person-centered care in long-term care facilities.	337 (240:97)	Mean: 81	In-person interview	Content analysis	Preferences for interpersonal interactions (greetings, staff showing care, and staff showing respect), coping strategies, personal care (bathroom needs, setting up bedding), and healthcare discussions. Specific qualities and characteristics of care interactions that are necessary to fully meet their everyday preferences
Harrison et al. (2017) [27]/USA	Qualitative study	To describe resident perspectives of resident-centered care (RCC) in 10 U.S. nursing homes in order to highlight the meaning of the term RCC	227 (Not reported)	52–101	Focus group interview	Phenomenological approach	RCC has meaning in ways that are consistent with in tentions at the national and state levels to advance culture change in nursing homes, including efforts to create a more homelike environment, increase resident decision making and direction of his or her lifestyle, and put residents first.
Scales et al. (2019) [28]/USA	Qualitative study	To elicit input from a range of nursing home stakeholders about how to enhance resident and family engagement in care planning and delivery with a view to developing an operational framework for PDCP (Person Directed Care Planning).	97 (87:10) Residents:16:3 Family members: 4:2 NA,LPN:13:0 Managers/administrators: 55:4	Residents: 53–92 Family members: 50–79 NA,LPN: 27–61 Managers/administrators: 25–65	Semi-structured interview	Qualitative data analysis	Strategies for supporting resident and family engagement in care planning The different roles that support resident and family engagement, and The perceived limits on achieving PDCP
Vennedey et al. (2020) [29]/Germany	Qualitative study	To analyse patients’ perspectives of facilitators and barriers towards implementing PCC.	25 (17:8)	Mean: 60	Semi-structured individual interview	Content analysis	Facilitators of and barriers to PCC were explained as microlevel, mesolevel, and macrolevel. The importance of being an active patient by taking individual responsibility for health was emphasized. Facilitators of PCC were functioning teams and healthy staff members, barriers to PCC were a lack of transparency in financing and reimbursement.
Sims-Gould et al. (2014) [30]/Canada	Larger mixed methods	To examine long-term care (LTC) resident and staff perceptions on the decision to use hip protectors and identify the factors that influence attitudes toward hip protector use	27 (19:8)	72–91 (mean: 87.8)	Focus group interview	Thematic analysis	Residents’ concerns with the physical aspects of hip protectors Residents’ assessment of their need (or lack thereof) for hip protectors Residents’ desire to be cooperative within the LTC environment
Kusmaul et al. (2020) [31]/USA	Descriptive qualitative study	To examine how people in different roles experienced choice and autonomy in four areas addressed by culture change.	32 (24:8)	Not reported	Semi-structured interview	Qualitative data analysis	It is difficult to balance resident choices with (a) the diverse needs/wants of other residents and (b) safety. Leaders, such as nurse managers, should provide ongoing education to residents, families, and staff to help negotiate these challenges.

F: Female; M: Male.

**Table 3 ijerph-19-08576-t003:** Synthesized themes of nursing home residents’ experience of person-centered care.

Key Concepts from First, Second-Order Constructs	Sub-Themes	Themes
Sharing daily life Consultation for residents’ concerns Conversation in positive language Expression in the words of the resident Extending the relationship to the family	1. Building closeness with nursing staff	I. Promotion of mutual understanding through communication
Getting information about the care provided and care plan Providing detailed historical information about residents Desiring medical providers with comprehensive medical knowledge Expecting a health care provider with a lot of knowledge about the residents	2. Sharing enough information about caring	
Nursing staff who are well aware of residents’ health conditions Providing care tailored to the needs of residents Wanting the assignment of a dedicated nursing staff Quick response of nursing staff to residents’ requests	3. Respect for the individuality of residents	II. Care that acknowledges the independence of residents
Able to live independently Self-determination ability Desire to maintain a hobby	4. Focusing on the residents’ own life	
Preventing residents’ body dysfunction Desire to maintain a healthy daily life Life without pain Monitoring of residents’ physical needs Positive attitude of nursing staff	5. Care to maintain physical and psychological well-being	III. Finding the optimized state
A place where their safety is guaranteed A comfortable space like home Environment where privacy is respected	6. Creating a therapeutic environment for care	

## Data Availability

Not applicable.

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
