# Peer review of "A Meta-Synthesis Study of Person-Centered Care Experience from the Perspective of Nursing Home Residents"

_ijerph, 2022, doi:10.3390/ijerph19148576_

Round 1
Reviewer 1 Report
Overall, this manuscript is quite satisfactory, and some suggestions are as follows:
- There should be a more detailed description and focus of the patient-centered care.
- The methods section should describe the research process in more detail, rather than the academic literature.
- Insufficient in the discussion section, such as the description of "supplier-centered" and "resident-centered".
- Specific limitations and recommendations for the findings of this manuscript should be made.
Reviewer 2 Report
The topic is interesting and the article is well organized. The structure of the article is presented clearly. However, I have some points:
1) Why the authors did not use ScienceDirect as a search engine? I used your keywords there and I had found more than 900 articles that need to be filtered and considered in the research.
2) The outcome and the results for the included articles should be displayed as a taxonomy figure to explain the distribution of the articles in a systematic layout. The taxonomy figure makes the search direction clear for the readers and opens future research directions.
Also, I suggest splitting the "Age of participant" column from Table 2 and making it a new figure. The age factor is very important to be highlighted in such studies.
Reviewer 3 Report
Thank you for the opportunity to engage with your work and the important topic of elderly care. The article is clearly structured and methodologically rigorous. I clearly recommend that it be published after some editing. I have the following comments which I hope will help you further on in your work.
1. in the introduction you cite only a small selection of literature, here you could usefully draw on a wider selection to show that many of the same issues exist across countries globally.
2. The focus on person centered care is relevant. I was curious about you deal with the history of the concept. You do not define the term in the introduction. Holistic patient-centered nursing has a long history in nursing, but person-centered care is a more recent concept. You include it as one of your MESH terms, but how have you defined it and how does it appear in your search. From the studies you have included it seems that it covers a range of caring practices, which is fine but this need to be clear. Do all the studies explicitly state that they are using person centered care or is it something you interpret that the care that goes on in these instances is person centered care. The term has a particular history, also as part of public policy, so important to focus on how you introduce it, where you draw the definition from and how the term appears in the studies.
3. results: you have studies from different countries with very different health systems and consequently different ways of organizing care for the elderly and NH. You don't write anything about this, but I would think it was important in order to understand where the elderly are talking from, what kind of care they have experienced and the level of care. A NH is in some places, a public offer, only for the most vulnerable elderly and in other places a private offer, which elderly can move into when they themselves think they want. The way you write it now it is as if NH and person centered care are universal standards/phenomena. More context here would provide a fuller analysis.
small things: line 20: the definition of nursing home is special, I would think it is primarily a place that provides care and support to older people who can no longer live alone. Here you could be more precise, what definitions of NH have you come across and just the fact that these definitions can be different is also something that helps to understand the differences there are.
line 44: higher level of analysis - what is meant by this, seems to imply better, but I think you mean provides a comparative or overview of related themes ?
line167: the resident says she doesn't want to go home - this statement make the reader wonder what kind of NH is being referred to in the particular study. In other NH the residents do not "go home"?
Language: I am not a native English speaker myself, but I recommend a proof reading by a native speaker, as I notice some irregular syntax when reading through the draft.
